# Direct estimation of fetal head circumference from ultrasound images based on regression CNN

**Jing Zhang**[1]                 JING.ZHANG@INSA-ROUEN.FR
**Caroline Petitjean**[1]            CAROLINE.PETITJEAN@UNIV-ROUEN.FR
**Pierre Lopez**[1]             PIERRE.LOPEZ@ETU.UNIV-ROUEN.FR
**Samia Ainouz**[1]             SAMIA.AINOUZ@INSA-ROUEN.FR
[1]*Normandie Univ, INSA Rouen, UNIROUEN, UNIHAVRE, LITIS, Rouen, France*

**Editors:** Accepted to MIDL 2020

## Abstract

The measurement of fetal head circumference (HC) is performed throughout the pregnancy as a key biometric to monitor fetus growth. This measurement is performed on ultrasound images, via the manual fitting of an ellipse. The operation is operator-dependent and as such prone to intra and inter-variability error. There have been attempts to design automated segmentation algorithms to segment fetal head, especially based on deep encoding-decoding architectures. In this paper, we depart from this idea and propose to leverage the ability of convolutional neural networks (CNN) to directly measure the head circumference, without having to resort to handcrafted features or manually labeled segmented images. The intuition behind this idea is that the CNN will learn itself to localize and identify the head contour. Our approach is experimented on the public HC18 dataset, that contains images of all trimesters of the pregnancy. We investigate various architectures and three losses suitable for regression. While room for improvement is left, encouraging results show that it might be possible in the future to directly estimate the HC - without the need for a large dataset of manually segmented ultrasound images. This approach might be extended to other applications where segmentation is just an intermediate step to the computation of biomarkers.
**Keywords:** CNN, deep regression, ultrasound images, fetus head circumference

## 1. Introduction

Automated measurement of fetal head circumference (HC) is performed throughout the pregnancy as a key biometric to monitor fetus growth and estimate gestational age. In clinical routine, this measurement is performed on ultrasound (US) images, via manually tracing of the skull contour or fitting it to an ellipse. Indeed, identifying the head contour is challenging due to low signal-to-noise ratio in US images, and also because the contours have fuzzy (and sometimes missing) borders (Fig. 1). Manual contouring is an operator-dependant operation, prone to intra and inter-variability, which provokes inaccurate measurements (Sarris et al., 2012). More precisely, the 95% limits of agreement are ±7mm for the intra-operator variability and ±12mm for the inter-operator variability (Sarris et al., 2012, Tab. 1 p. 272).

Some works have been proposed to automate the measurement of fetal head circumference in US images, such as (Li et al., 2017; Lu et al., 2005; Jardim and Figueiredo, 2005).

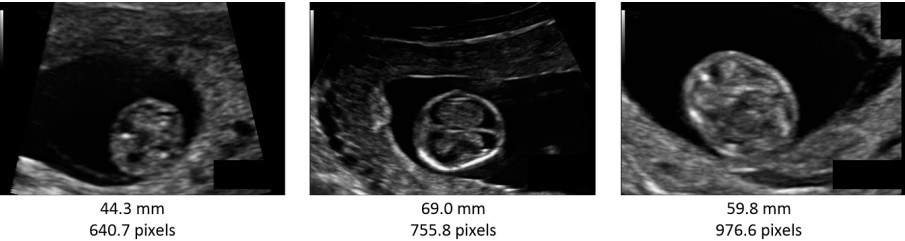

44.3 mm
640.7 pixels

69.0 mm
755.8 pixels

59.8 mm
976.6 pixels

Figure 1: Ultrasound images of fetal head from (van den Heuvel et al., 2018b). Corresponding head circumference (HC) is displayed in millimeters and pixels.

However, there are now more and more works aiming at directly extracting biomarkers from medical images, such as organ volume, area or features, to help clinical diagnosis. The goal is to avoid intermediate steps, such as segmentation, that maybe computationally expensive (both for model training and labeling) and prone to errors (Zhen and Li, 2015). For example, in (Zhen et al., 2015), the authors propose a learning-based approach to perform a direct volume estimation of the cardiac left and right ventricles from magnetic resonance images, without segmentation. The approach consists in computing shape descriptors using a bag-of-word model, and to perform Bayesian estimation and regression forests. By taking advantage of the power of convolutional neural networks (CNN), one can now skip the feature design step and learn the features, while at the same time performing the prediction of the value of interest, i.e. performing regression. Note that regression CNN have found several applications in the field of computer vision, such as head-pose estimation (Liu et al., 2016), facial landmark detection (Sun et al., 2013) and human-body pose estimation (Toshev and Szegedy, 2014).

In this work, we investigate if such a direct approach is reasonable to estimate the HC from ultrasound images, without having to resort to segmentation. Our approach is based on a regression CNN, for which we investigate four architectures, which differ by their complexity, and explore three losses for regression. Our experiments are carried out on the public dataset HC18 (van den Heuvel et al., 2018b). To our knowledge, this is the first attempt to directly assess the fetal head circumference, without resorting to segmentation.

The rest of the paper is organized as follows. Section 2 introduces related works about HC measurement in ultrasound images. Section 3 describes the proposed architecture and the loss functions. Experiments are conducted in Section 4. The conclusion and future works are drawn in Section 5.

## 2. Related works

Several approaches have been proposed in the literature to measure the head circumference in US images, based on image segmentation. Usually they follow at two-step approach, namely fetal head localization and segmentation refinement. In (van den Heuvel et al., 2018a), the first step consists in locating the fetal head via machine learning, with Haar-like features used to train a random forest classifier; and the second step consists in the measurement of the HC, via ellipse fitting and Hough transform. Similar method is used

in (Li et al., 2017). Other approaches build upon deep segmentation models also in a two-step process, prediction and ellipse fitting (Kim et al., 2019). In (Budd et al., 2019), the standard segmentation model U-Net (Ronneberger et al., 2015) is trained using manually labeled images, and segmentation results are fitted to ellipses. In (Sobhaninia et al., 2019), authors build upon the same idea, combining image segmentation and ellipse tuning together in a multi-task network.

## 3. CNN regressor

Standard CNN have several convolutional layers followed by fully-connected layers, ended with a classification softmax layer. Adapting a classification CNN architecture to regression consists in removing the softmax layer and replacing it by a fully connected regression layer with linear or sigmoid activation. Linear activation means that the transfer function is a straight line, thus the activation is proportional to input, and not confined to a specific range.

### 3.1. Model architectures

We have experimented four deep models with varying numbers of parameters and depths: two custom models and two common architectures. We have considered two simple models inspired by the base regressor of (Dubost et al., 2019): a first model called CNN_263K, with around 263K parameters and the second one called CNN_1M which has around 1M parameters (see Fig. 2). We also experimented VGG16 (+14M parameters) (Simonyan and Zisserman, 2015) and Resnet50 (+23M parameters) (He et al., 2016) pre-trained on ImageNet, and subsequently trained on our dataset. In each model, the fully connected regression layer has linear activation.

### 3.2. Regression loss function

Conventional regression loss functions are metrics-inspired losses, namely the Mean Absolute Error (MAE), Mean Squared Error (MSE) and Huber Loss (HL), defined as:

$$MAE = \frac{1}{n} \sum_{i=1}^{n} |p_i - g_i| \tag{1}$$

$$MSE = \frac{1}{n} \sum_{i=1}^{n} (p_i - g_i)^2 \tag{2}$$

$$HL = \begin{cases} \dfrac{1}{n} \sum_{i=1}^{n} \dfrac{1}{2}(p_i - g_i)^2, & \text{for} \quad |p_i - g_i| < \delta \\ \dfrac{1}{n} \sum_{i=1}^{n} \delta * (|p_i - g_i| - \dfrac{\delta}{2}), & \text{otherwise} \end{cases} \tag{3}$$

where predicted (resp. ground truth) values are denoted $p_i$ (resp. $g_i$). Huber loss is less sensitive to outliers than the quadratic loss (Esmaeili and Marvasti, 2019). As there is no heuristics to chose one loss over the other, we experience these three loss functions, as advocated in (Lathuilière et al., 2019).

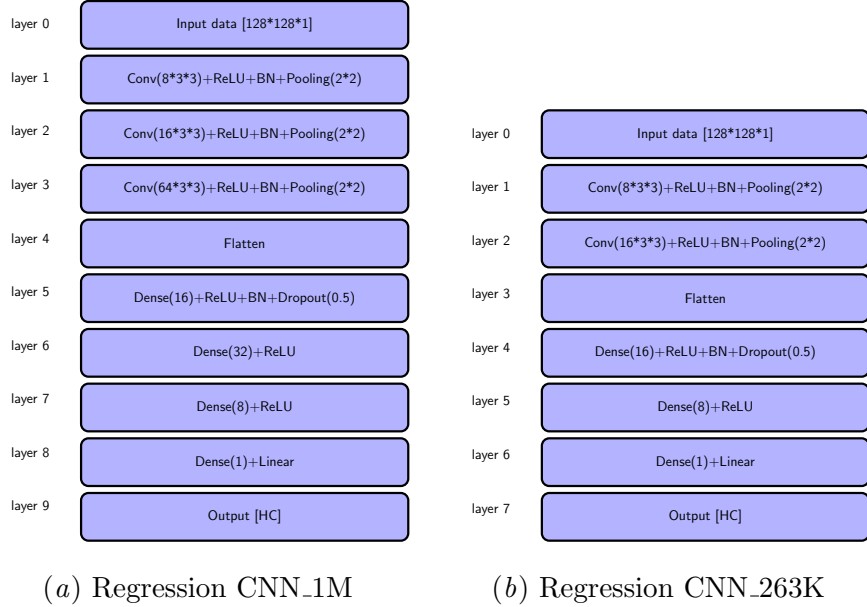

(a) Regression CNN_1M          (b) Regression CNN_263K

Figure 2: Architectures of the custom CNN based regression models

## 4. Experiments

### 4.1. The HC18 dataset

We use the HC18 *training* dataset (van den Heuvel et al., 2018b), that contains 999 US images acquired at varying times during the pregnancy, along with the corresponding head circumference[1]. HC values range from 439.1 pixels (44.3 mm) to 1786.5 pixels (346.4 mm), with average value being 1263.3±264.4pixels (174.4±65.2mm). We randomly split the dataset into a training (600), a validation (200) and a test set (199), except for the images that were made during one echographic examination, that are assigned the same set. We augment the data of the training set to 1800 images, by performing horizontal flipping, translation with 5 pixels offset, and rotation with 10 degrees.

Image preprocessing includes a resizing from $800 \times 540$ pixels to $224 \times 224$, and normalization by subtracting the mean and dividing by standard deviation. The HC values are normalized by dividing the maximum value of HC, in order to improve convergence.

### 4.2. Experimental setup

All the experiments are performed with 5-fold cross validation. The metrics to evaluate the results are Mean Absolute Error (mae) measured in pixels and in mm, and the percentage of mae (pmae). We have empirically set $\delta = 0.5$ in Huber loss. Models are trained with a batch size of 8, a learning rate of $1e^{-3}$, and Adam as optimizer. Models are implemented with Keras and TensorFlow.

---

1. The HC18 challenge is rather dedicated to head segmentation and evaluation on the HC18 *test* set requires to submit the parameters of an ellipse, which we do not have in our case.

## 4.3. Results

Table 1: Performance of regression models in terms of mean absolute error (mae) in pixels and %mae ($\pm$ standard deviation) for three different loss functions: MSE, MAE, HL

| | CNN_263K | | CNN_1M | | Reg-VGG16 | | Reg-ResNet50 | |
|---|---|---|---|---|---|---|---|---|
| loss | mae(pix) | pmae(%) | mae(pix) | pmae(%) | mae(pix) | pmae(%) | mae (pix) | pmae(%) |
| MSE | 90.18±86.42 | 8.74±12.51 | 50.96±58.61 | 4.96±7.85 | 38.85±40.31 | 5.31±5.63 | 36.21±35.82 | 4.62±4.27 |
| MAE | 101.85±108.51 | 10.99±18.48 | 51.61±59.96 | 5.15±8.66 | 40.17±40.99 | 5.26±5.79 | 37.34±37.46 | 4.85±4.93 |
| HL | 98.18±89.77 | 9.69±13.9 | 53.87±66.46 | 5.45±9.08 | 40.7±40.07 | 5.67±5.19 | 38.18±37.32 | 5.16±4.84 |

Table 2: Performance of Reg-Resnet50 vs Reg-VGG16 in terms of mae (pixels) ($\pm$ standard deviation) with and without data augmentation (DA).

| | Reg-Resnet50 | | Reg-VGG16 | |
|---|---|---|---|---|
| loss | without DA | with DA | without DA | with DA |
| MSE | 63.92±63.61 | 36.21±35.82 | 66.84±67.48 | 38.85±40.31 |
| MAE | 62.44±63.63 | 37.34±37.46 | 67.71±68.03 | 40.17±40.99 |
| HL | 66.62±66.18 | 38.18±37.32 | 67.02±76.08 | 40.7±40.07 |

Table 3: Performance of Reg-Resnet50 vs Reg-VGG16 in terms of mae (pixels and mm) for three different loss functions: MAE, MSE, HL with data augmentation. $^{\dagger}$: significantly different ($p < 0.05$) from all other methods, $^{\circ}$: significantly different ($p<0.05$) from all other methods, except for Reg-VGG16-MAE and Reg-VGG16-HL.

| | Reg Resnet50 | | Reg VGG16 | |
|---|---|---|---|---|
| loss | mae (pixels) | mae (mm) | mae (pixels) | mae (mm) |
| MSE | 36.21±35.82$^{\dagger}$ | 4.52±4.27$^{\dagger}$ | 38.85±40.31 | 4.87±5.81 |
| MAE | 37.34±37.46 | 4.78±4.41 | 40.17±40.99$^{\circ}$ | 5.46±5.99$^{\circ}$ |
| HL | 38.18±37.32 | 4.68±4.37 | 40.7±40.07$^{\circ}$ | 5.19±5.42$^{\circ}$ |

In our experiments, we compare the four regression models and the three loss functions, the Mean Absolute Error (MAE), the Mean Squared Error (MSE) and the Huber Loss (HL), and assess the added value of data augmentation.

Results in Tab. 1 show that the MSE loss obtains the best results, while the MAE and Huber loss have similar accuracy. Best results are obtained with Reg-VGG16 and Reg-ResNet50, which argues for a deeper architecture, with more power to grasp the image features. Reg-ResNet50 with MSE is found particularly powerful, as confirmed by the loss evolution during training and validation in Fig. 3.

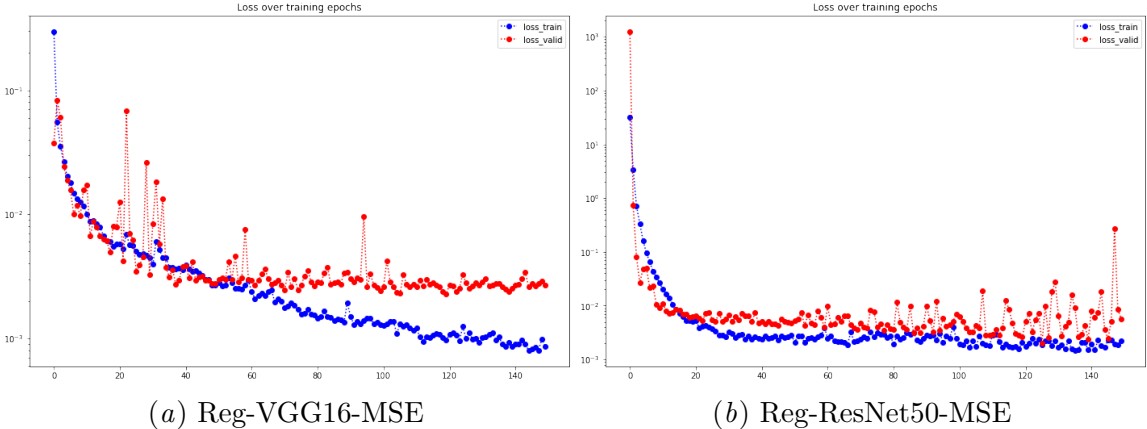

(*a*) Reg-VGG16-MSE    (*b*) Reg-ResNet50-MSE

Figure 3: Training and validation losses for Reg-VGG16 and ResNet50 with MSE

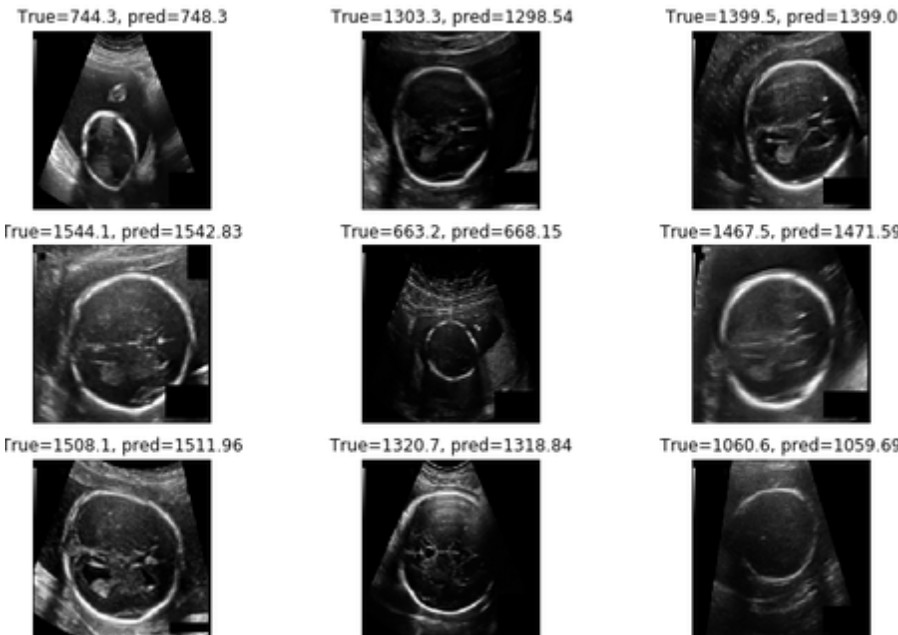

Figure 4: Samples of well predicted HC values (in pixels) with corresponding US images, with Reg-Resnet50-MSE

Thus, in the rest of the experiments, we focus on Reg-VGG16 and Reg-Resnet50 only. First, looking at the contribution of data augmentation, we can gather from Tab. 2 that data augmentation is really necessary to get a boost in performance, the error being divided by almost 2 with data augmentation. Then, we use a paired Wilcoxon signed-rank test to evaluate if the differences between methods and regression losses are significant: it appears that Reg-Resnet50 with the MSE loss has a significantly different error ($p < 0.05$) than the rest of the methods, and is thus the best setting in this case, as shown in Table 3, where

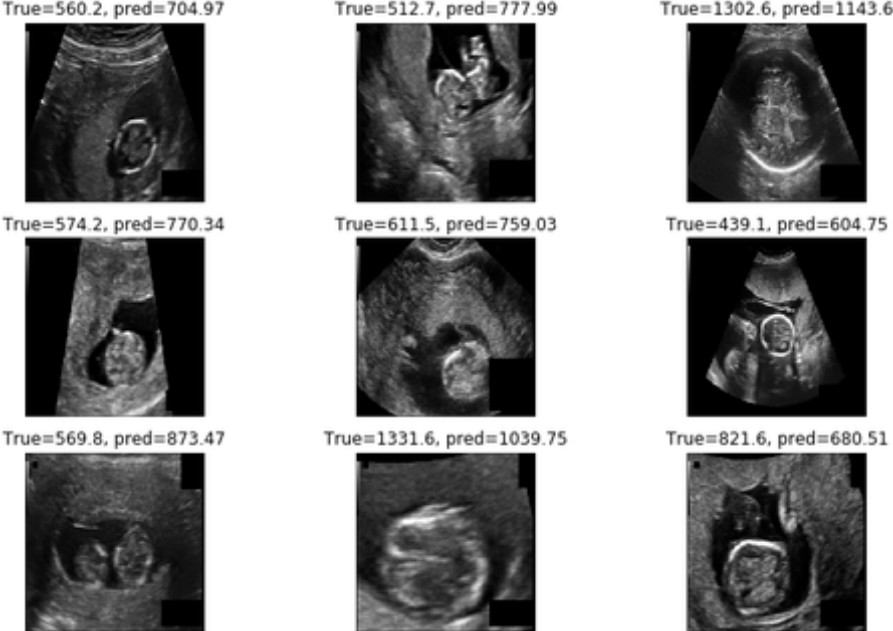

Figure 5: Samples of incorrectly predicted HC values (in pixels) with corresponding US images, with Reg-Resnet50-MSE

errors are reported in mm also. The best configuration thus has an error of 4.52±4.27 mm. This value is to be compared to the accuracy obtained by segmentation-based approaches: 2.12 ± 1.8 mm in (Sobhaninia et al., 2019), 2.8 ± 3.3 mm in (van den Heuvel et al., 2018a) and 1.81 ± 1.6 mm in (Budd et al., 2019). One should handle this comparison with care, since results have not been obtained on the same dataset and/or using the same protocol; however, we can say that the error obtained by the CNN regressor is doubled w.r.t that of segmentation-based approaches. Furthermore, standard deviation is high and remains to be investigated and compared to segmentation-based approaches.

The analysis of the prediction correctness w.r.t the images shows that correct predictions mainly stem from low speckle and highly contrasted US images; where the skull is rather correctly outlined, as shown in Figure 4, whereas images with a high level of speckle inside and outside the skull, that include other structures, yield high errors (Figure 5).

## 5. Conclusion

In this work, we have proposed an approach to directly estimate the fetal head circumference from US images by regression CNN. Our goal was to estimate how far a direct estimation method of the HC via regression was, from conventional prediction methods, which are based on segmentation and ellipse fitting. The rationale behind our approach is to remove the need for segmenting the US image. We compared several regression CNN architectures and three loss functions. Experimental results showed that the deeper model Reg-ResNet50 performed better, along with the MSE loss function. Encouraging results are obtained, since the best

models results in error comparable to manual measurement variability; however room for improvement for CNN-based regressors is left, especially when comparing to the accuracy of segmentation-based approaches. Future work will focus on designing the network so that the feature extraction is fostered in a way to segment the image - without segmentation ground truth. For this, we will investigate attention mechanisms and multi-task learning. We will also investigate whether errors are related to gestational age, as is the case for manual measurements (Sarris et al., 2012).

## Acknowledgments

The authors would like to thank the China Scholarship Council (CSC) for supporting Jing Zhang and acknowledge the CRIANN (Centre des Ressources Informatiques et Applications Numérique de Normandie, France) for providing computational resources.

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
