# OpenReview forum: "Direct estimation of  fetal head circumference from ultrasound images based on regression CNN"
_MIDL.io/2020/Conference — MIDL 2020_

### Official Review · AnonReviewer3 · 2020-02-20
**Interesting approach to determine the fetal head circumference**

**Rating:** 3
**Confidence:** 5
**Recommendation:** Oral, Poster

**Summary:**

The authors use a CNN to directly predict the fetal head circumference from 2D ultrasound images instead of first segmenting the fetal head circumference. They use the publicly available HC18 dataset and train two different CNN architectures using three different loss functions. The results in Table 1 and 2 do not show very good results, since the MAE is more than one cm, while other papers show results around 2mm. It was surprising for me to read 'By taking into account pixel size, ..., the mean absolute difference between predicted HC and ground truth HC in mm is 2.232mm (+/- 1.49)'

**Strengths:**

The authors are the first to directly determine the HC from an ultrasound image. The paper is well written and clearly describes two experiments. They experimented with three losses and show nice training graphs that explain their conclusions.

**Weaknesses:**

The results in Table 1 and 2 show a mean absolute error above 1cm, but the text reports an mean absolute difference of 2.216mm. I do not understand this difference.

The HC18 dataset also has an independent test set, which makes it possible to compare the results of different algorithms on the same dataset. It would be good to determine the Mean absolute difference in HC for this algorithm on the test set and report the results, so other authors can directly compare their results.

The authors mention that they perform flipping, translation and rotation, but do not mention the range (e.g. horizontal and vertical flipping? translation between ... and ... pixels/mm rotation of ... degrees). I also question if the rotation is a valid augmentation for ultrasound images, since the shadowing always occurs in the direction away from the transducer.



**Justification Of Rating:**

The authors show a novel approach to this problem and have written an easy to understand paper about this. It is a weak accept, because the authors need clarify if the results show a mean absolute difference of 2mm or 14mm.

**Paper Type:**

validation/application paper

**Questions To Address In The Rebuttal:**

The Tables should include explanation of the abbreviations and should also include the unit (probably mm).

The authors mention "data augmentation ... is really necessary to get a boost in performance", while Table 2 shows for the Huber Loss not much increase, so please explain this statement.

The introduction mentions: "several losses for regression", please adjust this to 'three losses for regression'


**Special Issue:**

no

---

> ### Author Response · Authors · 2020-03-27
> **Corrections of inconsistent results**
>
> We thank you for your valuable questions and suggestions. They have definitely helped us to improve our paper.
>
> > The results in Table 1 and 2 show a mean absolute error above 1cm, but the text reports a mean absolute difference of 2.216mm. I do not understand this difference.
> $\rightarrow$ You were absolutely right, this was not consistent. There were indeed mistakes in the tables, due to miscomputations with incorrect pixel size and image size, whose correct value is 224*224. This has led us to entirely rewrite the result tables.  We are sorry for that. We have now modified the paper with the correct results. It appears that now the best error is around 36±35pixels, which means 4.52±4.27 mm, obtained with ResNet50. The errors are larger than previously – however, we find it more consistent with the fact that segmentation-based approaches can reach a 2mm average error. The discrepancy shows between regression-based and segmentation-based method that a large room for improvement is left for direct estimation method. Note that this updated result is still of the same order of magnitude than inter-operator variability measurements: 95% limits of agreement are +/-7mm inter-operator variability and +/-12mm for intra- variability (Sarris et al, 2012).
>
> > It would be good to determine the Mean absolute difference in HC for this algorithm on the test set and report the results, so other authors can directly compare their results.
> $\rightarrow$ The HC values for the test images are not available. In order to evaluate results on the test set online, one must submit the parameters that describe an ellipse corresponding to the head contour, i.e.: the center coordinates, the semi-minor and major axis lengths, and the orientation angle – which unfortunately we cannot do since our method does not provide them, as the output is only the head circumference.
>
> > The authors mention that they perform flipping, translation and rotation, but do not mention the range (and horizontal or vertical flipping)?
> $\rightarrow$ We have added the range of the rotation (10°) / translation (5 pixels) and the flipping direction (horizontal) in the paper.
>
> > I also question if the rotation is a valid augmentation for ultrasound images, since the shadowing always occurs in the direction away from the transducer.
> $\rightarrow$ You are right. As you can see from the range, the rotation angle of 10° is not too large and allows for the shadowing to occurs always downward. However, we will consider the validity of data augmentation in future experiments.
>
> > The Tables should include explanation of the abbreviations and should also include the unit (probably mm).
> $\rightarrow$ We have now included this in the caption.
>
> > The authors mention "data augmentation ... is really necessary to get a boost in performance", while Table 2 shows for the Huber Loss not much increase, so please explain this statement.
> $\rightarrow$ Since the results have been modified, as explained above, we have found that the boost of performance with data augmentation is about the same for all 3 loss functions. The sentence has thus been removed.
>
> > The introduction mentions: "several losses for regression", please adjust this to 'three losses for regression
> $\rightarrow$ We adjusted this.
>
> We would like to inform you that, following some questions of the reviewers, we have added additional results with VGG16 and ResNet50 and statistical tests.

---

### Official Review · AnonReviewer1 · 2020-03-10
**Regression CNNs for head circumference from ultrasound.**

**Rating:** 3
**Confidence:** 4
**Recommendation:** Poster

**Summary:**

The authors compare two regression CNNs for estimation of fetal head circumference (HC) in ultrasound images. This is an useful and important application as head circumference is a key measurement for monitoring of fetal growth and estimation of gestational age. The paper describes two architectures to directly estimate the HC without using manually labeled data.

**Strengths:**

The direct estimation of HC without using manually labeled data is interesting. Automatic measurements in ultrasound images are important as these are often challenging to interpret and inter/intra-observer variability is high. For measurements related to growth, like in this application, it is particularly important to have high precision measurements.

**Weaknesses:**

Some points could be clarified:
- The motivation for the two architectures presented.
- Does the data contain 999 unique foetuses or 999 unique acuisitions?
- What is the accuracy required by the clinical users for this application?
- Is the error independent of HC? In other words, does the method perform equally well on early stage and late stage foetuses?
- The authors claim "small labelling cost". This could be commented in the discussion.



**Detailed Comments:**

- In section3: "fully connected regression linear with linear or sigmoid activation" should be "fully connected regression layer..."
- Define all abbreviations the first time.
- Define DA
- Include unit for MAE in tables



**Justification Of Rating:**

Interesting method although the motivation for the actual architectures is not well explained. The paper describes and important and well-defined application where automated measurements could have a great impact.

**Paper Type:**

both

**Questions To Address In The Rebuttal:**

- Motivation for the architectures used
- The required accuracy fo this particular application

**Special Issue:**

no

---

> ### Author Response · Authors · 2020-03-27
> **Splitting for train/validation/test set explained**
>
> We thank you for your valuable questions and suggestions. They have definitely helped us to improve our paper.
>
> > The motivation for the two architectures presented.
> $\rightarrow$ We have experimented with four deep models with varying numbers of parameters and depths: two custom models and two common architectures. We have considered two simple models inspired by the base regressor of (Dubost et al MICCAI 2019 : Hydranet: Data Augmentation for Regression Neural Networks), a first model called CNN_263K, with around 263 000 parameters and the second one called CNN_1M which has around 1M parameters. We also experimented  VGG16  (+14M parameters) and Resnet50 (+23M parameters) pre-trained on ImageNet. These two models are then trained on our dataset. We have added this information in the paper.
>
> > Does the data contain 999 unique foetuses or 999 unique acquisitions?
> $\rightarrow$ In the HC18 dataset, most of the ultrasound images come from single echoscopic examinations. However, a small part of the images was made during the same examinations and have therefore a very similar appearance.  We paid attention that all images that were made during one echographic examination were assigned to either the training, the validation or the test. Apart from that, images were randomly distributed. We have added this important information in the paper.
>
> > What is the accuracy required by the clinical users for this application?
> $\rightarrow$ The clinical required accuracy is related to the intra- and inter-expert variability of manual measurements. If we refer to (Sarris et al, 2012), we can see from their study on 175 scans, that the 95% limits of agreement are +/-7mm inter-operator variability and +/-12mm for intra- variability. In (Heuvel et al, 2019), the mean absolute error between two operators is given, ranging from 1.8+/-1.5 mm for 1st-trimester measurement to 5.4+/-4.6 mm for 3rd-trimester measurements. This information is in the paper.
>
> > Is the error independent of HC? In other words, does the method perform equally well on early stage and late stage foetuses?
> $\rightarrow$  This is an interesting question; indeed, intra- and inter-operator variability errors increase with the fetus age (Sarris et al 2012). However we don’t have the gestational age information in this dataset; but we could very well draw some Bland-Altman plot, i.e. we could draw the difference between predicted and real values, w.r.t. to the real values. This idea is definitely worth investigating in future works.
>
> > The authors claim "small labeling cost". This could be commented in the discussion.
> $\rightarrow$ By labeling cost we meant that, in order to train a segmentation network, a large dataset of labeled, i.e. manually segmented images is required. Manual contouring or labeling of medical images is generally tedious and lengthy. The interest of an approach that is directly estimating the value is that it requires only the corresponding HC value to train the regression network, and it does not require the costly step of manual contouring. We have reworded the sentence to clear its meaning.
>
> > In section3: "fully connected regression linear with linear or sigmoid activation" should be "fully connected regression layer..." - Define all abbreviations the first time. - Define DA - Include unit for MAE in tables
> $\rightarrow$ Thank you for these detailed comments. We have corrected these mistakes.
>
> We would like to inform you that, following some questions of the reviewers, we have added additional results with VGG16 and ResNet50 and statistical tests. As pointed out by one of the reviewer (4), there were some inconsistencies in the results. There were mistakes in the tables, due to miscomputations with incorrect pixel size and image size, whose correct value is 224*224. This has led us to entirely rewrite the result tables.  We are sorry for that. We have now modified the paper with the correct results. It appears that now the best error is around 36±35pixels, which means 4.52±4.27 mm, obtained with ResNet50. The errors are larger than previously – however, we find it more consistent with the fact that segmentation-based approaches can reach a 2mm average error. The discrepancy shows between regression-based and segmentation-based method that a large room for improvement is left for direct estimation method. Note that this updated result is still is of the same order of magnitude than inter-operator variability measurements: 95% limits of agreement are +/-7mm inter-operator variability and +/-12mm for intra- variability (Sarris et al, 2012).

---

### Official Review · AnonReviewer4 · 2020-03-14
**Not clearly written, results not ready for presentation**

**Rating:** 2
**Confidence:** 4

**Summary:**

The proposed problem, measuring head circumference from fetal ultrasound images, is very interesting and important.  It would be great to find an efficient and accurate way of solving it. The goal is to establish head circumference, a significant key feature in tracking development, without manual delineation or (automated) full brain segmentation.

**Strengths:**

Providing robust quantitative estimates for head circumference from ultrasound images addresses an important question.  THE HC18 dataset is used, providing a large training data set. Quantitative results from the literature are compared when evaluating performance.

**Weaknesses:**

The submission is moderately poorly written, which makes reading it hard.

The assembly of the proposed network/pipeline seems somewhat random and there is not much insight provided to the reader in the way of explanation. The authors use regression CNN, comparing two different types of architectures and several loss functions: CNN_1M and CNN_263K: What is the justification for these two systems? How did the authors come up with them?

The current performance comparison is very approximate. Do the baseline segmentation-based approaches (used as performance comparison) not have any open-source implementation to try on the current data sets? Is there an std value accompanying the mean value from the literature?

What are the low labeling costs mentioned for the new pipeline? Do those refer to the HC measures? Please, describe in more detail.

**Detailed Comments:**

Is image resizing cropping or rescaling?
Are the results statistically significantly different?

Future work: why multi-task learning?

Minor
=====
intra and inter- --> intra- ad inter-
directly extract biomarkers --> directly extracting biomarkers
feal head --> fetal head
we experience these loss functions --> we experiment with these loss functions
results ... shows --> results ... show
at varying time during the pregnancy --> at varying times of pregnancy
backbones -- colloquialism

**Justification Of Rating:**

The methodology is poorly explained and it is not clear what the main conclusion of the paper is. The results are below the segmentation techniques from the literature and have a higher variance than manual segmentation results.


**Paper Type:**

validation/application paper

**Special Issue:**

no

---

> ### Author Response · Authors · 2020-03-27
> **Writing improved + Justification of the architecture**
>
> We thank you for your valuable questions and suggestions. They have definitely helped us to improve our paper.
>
> > The submission is moderately poorly written, which makes reading it hard.
> $\rightarrow$ We corrected the paper as best as we could and tried to improve the writing.
>
> > CNN_1M and CNN_263K: What is the justification for these two systems? How did the authors come up with them?
> $\rightarrow$ We have experimented with four deep models with varying numbers of parameters and depths: two custom models and two common architectures. We have considered two simple models inspired by the base regressor of (Dubost et al MICCAI 2019 : Hydranet: Data Augmentation for Regression Neural Networks), a first model called CNN_263K, with around 263 000 parameters and the second one called CNN_1M which has around 1M parameters. We also experimented  VGG16  (+14M parameters) and Resnet50 (+23M parameters) pre-trained on ImageNet. These two models are then trained on our dataset. We have added this information in the paper.
>
> > Do the baseline segmentation-based approaches not have any open-source implementation to try on the current data sets?
> $\rightarrow$ We have not been able to find their implementation; however it should not be too difficult to implement a state of the art segmentation network such as U-Net and obtain some results on the HC18 dataset. This would indeed be the best way to guarantee a fair comparison between segmentation-based approach and direct estimation approach, on the same dataset and partition. This is indeed a work to be done in the near future.
>
> > Is there an std value accompanying the mean value from the literature?
> $\rightarrow$ Yes, you are right and we have now added the standard deviation:  2.12±1.8 mm in (Sobhaninia et al., 2019), 2.8±3.3 mm in (van den Heuvel et al., 2018a) and 1.81±1.6 mm in (Budd et al., 2019).
>
> > What are the low labeling costs mentioned for the new pipeline? Do those refer to the HC measures?
> $\rightarrow$ By labeling cost we meant that, in order to train a segmentation network, a large dataset of labeled, i.e. manually segmented images is required. Manual contouring or labeling of medical images is generally tedious and lengthy. The interest of an approach that is directly estimating the value is that it requires only the corresponding HC value to train the regression network, and it does not require the costly step of manual contouring. We have reworded the sentence to clear its meaning.
>
> > Is image resizing cropping or rescaling?
> $\rightarrow$ For the dataset, the images are resized to 224*224 by rescaling, not cropping.
>
> > Are the results statistically significantly different?
> $\rightarrow$ Now we have added statistical tests. We use a Wilcoxon signed-rank test to evaluate the differences between methods.
>
> > Future work: why multi-task learning?
> $\rightarrow$ The idea is to perform an auxiliary task that can help the primary task, in our case the prediction of HC values. In deep learning, multi-task learning is done in sharing parameters in some of the layers. Our idea is to simultaneously segment (in an unsupervised way) the image while at the same time learning to predict the value.
>
> We thank the reviewer for all the typos (s)he pointed out. We corrected them.
>
> We would like to inform you that, following some questions of the reviewers, we have added additional results with VGG16 and ResNet50. As pointed out by one of the reviewer (4), there were some inconsistencies in the results. There were mistakes in the tables, due to miscomputations with incorrect pixel size and image size, whose correct value is 224*224. This has led us to entirely rewrite the result tables.  We are sorry for that. We have now modified the paper with the correct results. It appears that now the best error is around 36±35pixels, which means 4.52±4.27 mm, obtained with ResNet50. The errors are larger than previously – however, we find it more consistent with the fact that segmentation-based approaches can reach a 2mm average error. The discrepancy shows between regression-based and segmentation-based method that a large room for improvement is left for direct estimation method. Note that this updated result is still is of the same order of magnitude than inter-operator variability measurements: 95% limits of agreement are +/-7mm inter-operator variability and +/-12mm for intra- variability (Sarris et al, 2012).

---

### Official Review · AnonReviewer2 · 2020-03-16
**Interesting paper, but limited results**

**Rating:** 3
**Confidence:** 4

**Summary:**

This paper is sort of a "negative results" paper. It is about testing a different (automatic and direct) approach to baby head circumference measurement in ultrasound, a very common procedure done in every pregnancy. Innovation in this procedure would have a significant impact on clinical practice. However, the authors found that their approach was not accurate enough. Probably others have also tried this approach, but not published it due to large errors in the measurement.

**Strengths:**

Using a public dataset and describing methods clearly, the authors have made an exemplary effort to publishing reproducible research.
The language reads well, the figures are informative and support the text nicely.
The background literature is well-presented.

**Weaknesses:**

By far the main weakness of the paper is that the results are far from applicable in routine clinical applications. We don't know if this is because the approach chosen by the authors would never work in practice, or because it just needs more fine-tuning or better training data/parameters.
Figure 2 could be made more useful by including more parameters. State the size of the conv layers in each layer (not just the input) and/or state explicitly the resize factor of the pooling layers. Linear activation is very uncommon, needs to be better specified.
The authors say that their future work will include VGG and ResNet testing. However, these networks are readily available in all deep learning environments. It should only take a few hours to test them on the given dataset. Why did you not test them already? Why wait for future research to do a task that should fit in maximum one day?

**Justification Of Rating:**

The topic and the clear presentation warrants acceptance. However, my concerns listed under the "Weaknesses" section are numerous, so I don't feel particularly strong about this paper. Maybe the authors could make improvements to make a stronger case for acceptance.

**Paper Type:**

both

**Questions To Address In The Rebuttal:**

If this paper will be published after revision, then I have a few suggestions to make it better:
- Make Figure 2 more informative
- In section 4.1, describe how you split the data into training and testing sets. Especially how did you choose patients whose data would go to the testing set. I hope the images were not randomized so one patient could be in both groups by different images.
- Define all acronyms when you first use them. Even CNN, and especially DA.


**Special Issue:**

no

---

> ### Author Response · Authors · 2020-03-27
> **Additional results from VGG16+ResNet50**
>
> We thank you for your valuable questions and suggestions. They have definitely helped us to improve our paper.
>
> > By far the main weakness of the paper is that the results are far from applicable in routine clinical applications.
> $\rightarrow$ We understand the reviewer’s concern. Actually, our goal was rather to estimate how far a direct estimation method of the HC via regression was, from methods typically used in clinical routine or segmentation-based methods. By establishing a baseline with this paper, regarding the performance of direct estimation of HC, we can now move further and propose improvement to the current approach. We have tried to make this clearer in the paper.
>
> > Describe how you split the data into training and testing sets.
> $\rightarrow$ In the HC18 dataset, most of the ultrasound images come from single echoscopic examinations. However, a small part of the images was made during the same examinations and have therefore a very similar appearance.  We paid attention that all images that were made during one echographic examination were assigned to either the training, the validation or the test. Apart from that, images were randomly distributed. We have added this important information in the paper.
>
> > The authors say that their future work will include VGG and ResNet testing. Why did you not test them already?
> $\rightarrow$ We actually have been able to run VGG16 and ResNet50 on the dataset in the meantime. We have updated the paper with the results since we had a bit of space left. Results on VGG16 and ResNet50 are actually better than the other 2 architectures.
>
> > Linear activation is very uncommon, needs to be better specified.
> $\rightarrow$ Linear activation means that the transfer function is a straight line, thus the activation is proportional to input, and not confined to a specific range. We added this definition to the paper.
>
> $\rightarrow$ We also improved Figure 2 and define acronyms as rightfully suggested.
>
> We would like to inform you that, following some questions of the reviewers, we have also added statistical tests in the new version of the paper. As pointed out by one of the reviewer (4), there were some inconsistencies in the results. There were mistakes in the tables, due to miscomputations with incorrect pixel size and image size, whose correct value is 224*224. This has led us to entirely rewrite the result tables.  We are sorry for that. We have now modified the paper with the correct results. It appears that now the best error is around 36±35pixels, which means 4.52±4.27 mm, obtained with ResNet50. The errors are larger than previously – however, we find it more consistent with the fact that segmentation-based approaches can reach a 2mm average error. The discrepancy shows between regression-based and segmentation-based method that a large room for improvement is left for direct estimation method. Note that this updated result is still is of the same order of magnitude than inter-operator variability measurements: 95% limits of agreement are +/-7mm inter-operator variability and +/-12mm for intra- variability (Sarris et al, 2012).

---

### Author Response · Authors · 2020-03-27
**Thank you to the reviewers**

We would like to thank the reviewers for their time. Their comments and questions have been thoroughly useful for us to improve the paper. We acknowledge that in our initial paper there were some important piece of information missing and some confusion. We hope to have cleared them up in our replies.

---

### Meta-Review · Area_Chair1 · 2020-04-06
**MetaReview of Paper222 by AreaChair1**

**Rating:** 3
**Recommendation For Accepted Papers:** Poster

**Metareview:**

The authors proposed to use a regression CNN to obtain a direct estimation of fetal head circumference from US images. The reviewers agree on the weak acceptance of this paper.


**Paper Type:**

validation/application paper

**Special Issue:**

no

---

### Decision · Program_Chairs · 2020-04-11

Accept